# Cell death and barrier disruption by clinically used iodine concentrations

Anne Steins[1,2], Christina Carroll[2], Fui Jiun Choong[1], Amee J George[1,3], Jin-Shu He[4], Kate M Parsons[4], Shouya Feng[4], Si Ming Man[4], Cathelijne Kam[1], Lex M van Loon[1,2], Perlita Poh[1], Rita Ferreira[1], Graham J Mann[1,2], Russell L Gruen[2], Katherine M Hannan[1,2], Ross D Hannan[1,2], Klaus-Martin Schulte[1,2,5]

Povidone-iodine (PVP-I) inactivates a broad range of pathogens. Despite its widespread use over decades, the safety of PVP-I remains controversial. Its extended use in the current SARS-CoV-2 virus pandemic urges the need to clarify safety features of PVP-I on a cellular level. Our investigation in epithelial, mesothelial, endothelial, and innate immune cells revealed that the toxicity of PVP-I is caused by diatomic iodine ($I_2$), which is rapidly released from PVP-I to fuel organic halogenation with fast first-order kinetics. Eukaryotic toxicity manifests at below clinically used concentrations with a threshold of 0.1% PVP-I (wt/vol), equalling 1 mM of total available $I_2$. Above this threshold, membrane disruption, loss of mitochondrial membrane potential, and abolition of oxidative phosphorylation induce a rapid form of cell death we propose to term iodoptosis. Furthermore, PVP-I attacks lipid rafts, leading to the failure of tight junctions and thereby compromising the barrier functions of surface-lining cells. Thus, the therapeutic window of PVP-I is considerably narrower than commonly believed. Our findings urge the reappraisal of PVP-I in clinical practice to avert unwarranted toxicity whilst safeguarding its benefits.

## Introduction

Elemental iodine ($I_2$) is an antimicrobial, which kills bacteria, viruses, and fungi at concentrations of a few parts per million (ppm) (Bartlett & Schmidt, 1957; Kelly, 1961; Gottardi, 1985). Polymer carriers such as polyvinylpyrrolidone (PVP) bind $I_2$ and are required to keep it stable in aqueous solutions (Shelanski & Shelanski, 1956; Pollack & Iny, 1985). Moreover, binding of $I_2$ by PVP is widely believed to afford slow release of $I_2$, thereby preventing toxicity to exposed tissues (Hickey et al, 1997) whilst preserving microbicidal effects of $I_2$ (Sauerbrei, 2020). As such, at 10% (wt/vol water) povidone-iodine (PVP-I), corresponding to 39.4 mmol/l of total iodine, free $I_2$ concentrations reach only 10 $\mu$mol/l or 2.54 ppm (Gottardi, 1985; Zamora, 1986; Hickey et al, 1997; Nagatake et al, 2002; Maurya et al, 2022). The widespread use of PVP-I in control and management of infectious disease (Eggers, 2019) comprises internal use on pharyngeal, intestinal, peritoneal, vaginal, ophthalmic, and joint surfaces, and external applications such as surgical scrub, wound care, and burns (Gilmore et al, 1978; Sindelar & Mason, 1979; Kahrom et al, 2017; Ruder & Springer, 2017; Roeckner et al, 2019; Tan & Johari, 2021). PVP-I is used in up to a third of septic abdominal surgeries (Whiteside et al, 2005). Extensive antiseptic, prophylactic, therapeutic, and patient-directed use is reflected by annual sales reaching 0.2 billion USD by 2030 (TMR, 2022). Recently, the use of PVP-I has been proposed to prevent community transmission of the SARS-CoV-2 virus via reduction in viral load in the upper aerodigestive tract with nasal spray and mouth gargle (Guenezan et al, 2021; Arefin et al, 2022; Lim et al, 2022).

Despite the long-time and widespread use of PVP-I, there is no consensus on its safety. Praise of iodine as a panacea for infectious disease meets hesitation considering its toxicity. To this end, a deeper understanding of PVP-I is needed to determine the effects it has on human cells and the mechanism by which it acts. We explored PVP-I toxicity with a focus on barrier integrity, investigating cells that line commonly exposed internal surfaces, including endothelial, epithelial, mesothelial, and immune cells. PVP-I caused rapid eukaryotic cell death by disruption of the cell membrane and mitochondrial membrane potential at concentrations 10- to 100-fold below those that can be reached in clinical use (Tan & Johari, 2021). Attack on lipid rafts disrupted tight junctions, opening tissue barriers to free diffusion. These mechanisms explain long-known local and systemic toxicities (Glick et al, 1985; Manfro et al, 2002; Ramaswamykanive et al, 2011; Song et al, 2018). The therapeutic window of PVP-I is far narrower than commonly believed. Below a threshold concentration of 0.1% PVP-I (wt/vol), short-term exposure and mid-term exposure are free of said eukaryotic toxicity, offering opportunities for its safe use.

[1]Division of Genome Sciences and Cancer, The John Curtin School of Medical Research, Australian National University, Acton, Australia  [2]College of Health and Medicine, Australian National University, Acton, Australia  [3]ANU Centre for Therapeutic Discovery, Australian National University, Acton, Australia  [4]Division of Immunology and Infectious Disease, The John Curtin School of Medical Research, Australian National University, Acton, Australia  [5]Department of Endocrine Surgery, King's College Hospital NHS Foundation Trust, London, UK

Correspondence: km.schulte@anu.edu.au
Deceased author: Katherine M Hannan died on March 10, 2022

# Results and Discussion

### Clinical concentrations of PVP-I are toxic to commonly exposed internal cell surfaces

Depending on the application, PVP-I is clinically used in a concentration range from 0.001% to 10% (wt/vol) (Tan & Johari, 2021). To determine the cellular toxicity of PVP-I across this range, endothelial (HMEC-1), mesothelial (LP-9), epithelial (A549), and immune cells (RAW264.7) were exposed to short-term (ST, 5 min) treatment with a range of PVP-I doses. At concentrations above 0.1% PVP-I, we observe that cellular metabolism and mitochondrial metabolism were abolished (Fig 1A and B) in a concentration-dependent fashion, similar to that previously observed in osteoblasts (Ede et al, 2016; Liu et al, 2017). When cells were exposed to stable solutions of PVP carrier alone or $I_2$, toxicity was solely caused by $I_2$ (Figs 1C–E and S1A–C). Above a threshold of free $I_2$ at 0.5 mM, elemental iodine is an irreversible respiratory poison, likely explaining topical toxicity such as retinal degradation (Shimada et al, 2020), and clinical sudden cardiovascular and metabolic collapse upon high-level systemic exposure (Glick et al, 1985; Kanakiriya et al, 2003; Ramaswamykanive et al, 2011), confirmed in animal models (Lores et al, 1981; Song et al, 2018). To translate these findings to a 3D environment, calcein AM–labelled epithelial spheroids were exposed to PVP-I or $I_2$. Within the first 3 min of exposure, the viability of cells on the peripheral surface of the spheroid was reduced upon contact with PVP-I or $I_2$ (Fig 1F and G). Within the next 30 min of exposure, the viability of cells within the core of the spheroids gradually reduces, presumably because of the diffusion of $I_2$ through the cellular structure (Fig S1D and E). Together, we showed that PVP-I is only safe below a concentration of 0.1% wt/vol. Above this threshold, PVP-I kills human cells where it might extend deep into exposed tissues.

### Fast release of $I_2$ from PVP-I causes toxicity

Interestingly, in the examined cell lines the $IC_{50}$ values of $I_2$ corresponded to the predicted $IC_{50}$ of PVP-I if all $I_2$ would be unbound to PVP (Table 1). Subsequent determination of the $I_2$ content in PVP-I and freshly prepared $I_2$ solution using UV-Vis spectrophotometry revealed that indeed similar concentrations of $I_2$ are present in both solutions at the $IC_{50}$ value of the cell lines, where 0.05% PVP-I equals 0.455 ± 0.061 mM $I_2$ (Tables 2 and 3 and Fig S2A and B). This challenges the proposed slow $I_2$ release system of PVP-I. To determine whether free $I_2$ (0.253 kD) or $I_2$ bound to PVP (40 kD) was responsible for the observed toxicity, PVP-I was filtered over a 10-kD column. Only the PVP-bound $I_2$ fraction retained above the filter induced cellular toxicity (Fig 1H and I), whereas the concentration of unbound and filterable $I_2$ of the PVP-I solution was non-toxic. This was validated by determination of the $I_2$ content in both fractions using UV-Vis spectrophotometry. This revealed all $I_2$ is found in the fraction retained above the filter, whereas no $I_2$ was detected in the filtered solution (Table 4 and Fig S2C). As PVP did not affect cell viability (Fig 1D and E), this suggested all PVP-bound $I_2$ is rapidly released when exposed to cells. Moreover, $I_2$ was inactivated by glutathione (GSH; Fig S2D), which is a tripeptide and protective

antioxidant found in very high concentrations in most human cells (Kerksick & Willoughby, 2005). This occurred in a 1:1 molecular ratio where GSH-inactivated $I_2$ was unable to induce cellular toxicity, supporting that first-order kinetic halogenation of $I_2$ with cellular content occurs (Truesdale & Luther, 1995).

Our observations challenge the entrenched dogma that PVP-based "*germicides containing a high level of molecular iodine are not irritating or toxic*" (Hickey et al, 1997). Previous work proposes slow release from PVP and fast dissociation of $I_2$ in pure aqueous media, thereby reducing its toxicity (Zamora, 1986). Other studies investigated the chemical behaviour of different $I_2$ solutions and scrutinized methods to differentiate $I_2$ species occurring in such solutions (Gottardi, 1985, 1999; Gottardi & Nagl, 2019). We propose that the toxicity of $I_2$-storing polymers is not dependent on the concentration of $I_2$ liberated at any particular moment. Rather, toxicity depends on the total amount of $I_2$, which can be liberated. This is due to the fact that the liberation of $I_2$ from such polymers is more or less instant as $I_2$ out reacts with halogen targets as soon as liberated from storage, thereby affording a near-instantaneous halogen challenge to any available targets.

### Prolonged exposure to low-dose PVP-I does not affect key cell functions

The broad pathogen inactivation by PVP-I (Eggers et al, 2018; Anderson et al, 2020; Sauerbrei, 2020; Tan & Johari, 2021) can be enhanced by prolonged exposure at low concentrations, that is, low-toxicity (LT) treatment (Hosseini et al, 2012). Bactericidal efficacy of PVP-I solutions is observed in the dilution range of 0.005–10%, with free $I_2$ concentrations peaking at a concentration of 0.1%. Former authors found only a range between 0.005% and 0.1% effective, as it caused only minor membrane damage (Van den Broek et al, 1982). Others found the safe concentration range to vary with cell type, such as <0.033% for the retina (Shimada et al, 2020), <0.05% for corneal endothelial cells (Jiang et al, 2009), and <0.05% for corneal epithelial cells (Swift et al, 2020).

Therefore, we explored the direct and prolonged toxic effects at concentrations ~10-fold below the $IC_{50}$ (Table 1) of PVP-I (i.e. 0.01%) and $I_2$ (i.e. 50 $\mu$M) for 30 min. Cell cycle progression of fast cycling immortalized endothelial and epithelial cells was not affected by PVP-I at 48 h after treatment (Figs 2A and S3). Slow cycling untransformed primary mesothelial cells had some increased retention in the G2/M phase, suggesting interference of PVP-I with mitotic division on long-term exposure (Figs 2A and S3). Mitochondrial oxygen consumption rate (OCR) was unaltered after treatment (Fig 2B). Evaluation of immune toxicity in freshly isolated murine leucocytes found no changes in cytokine secretion and leucocyte activation status after prolonged low-dose treatment (Fig 2C). Together, these findings suggest that LT treatment does not affect viability or functionality of human cells.

### Clinical concentrations of PVP-I rapidly disrupt the integrity of cell membranes

Investigating temporospatial events preceding cell death caused by 1% PVP-I, live-cell imaging of calcein AM–pre-labelled cells revealed that 90% of cells died within the first 5 s after exposure to

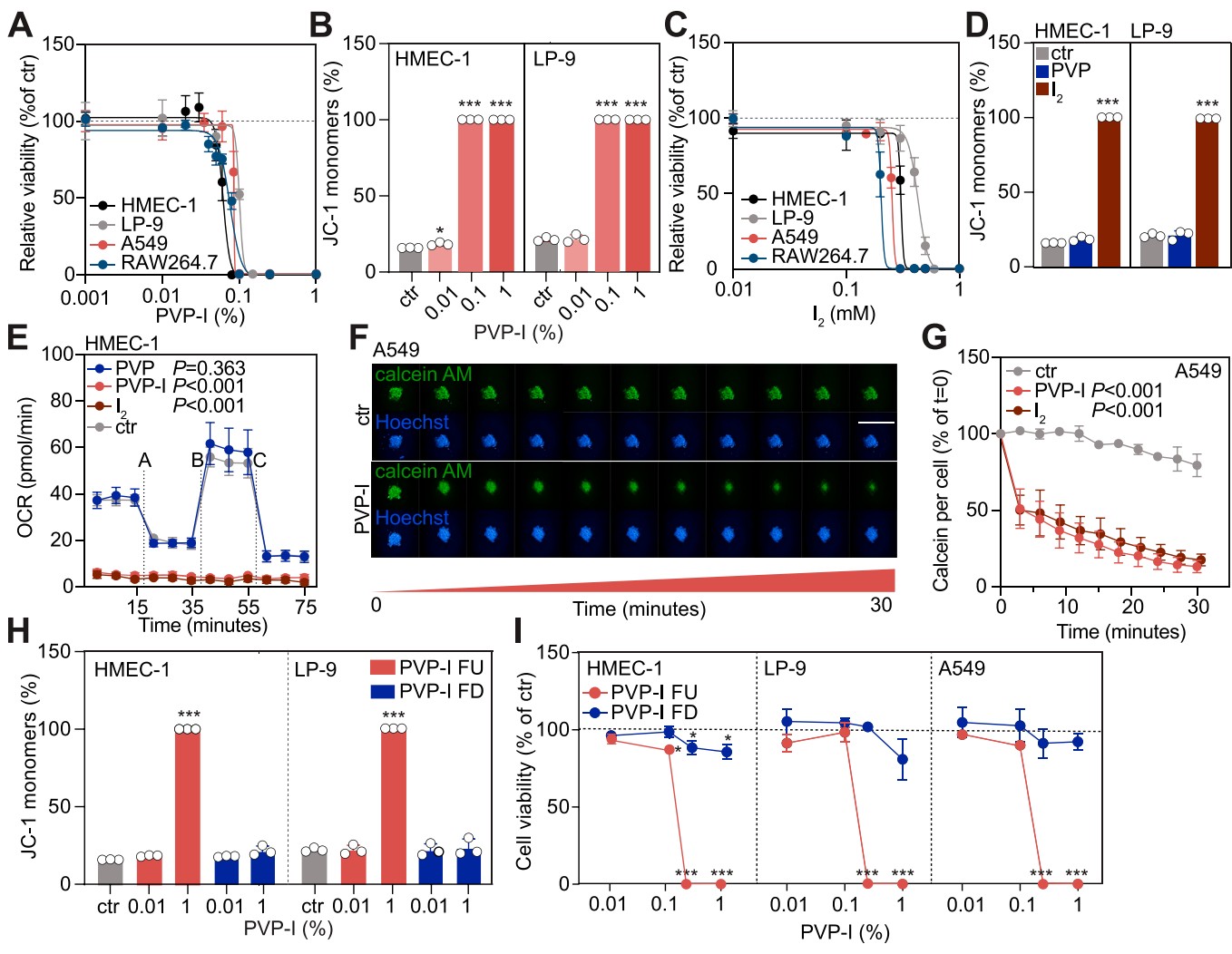

**Figure 1. Short-term exposure to below clinically used concentrations of PVP-I impairs the viability of various cell types.**
Cells were exposed to short-term (ST, 5 min) treatment with either control, 1% PVP, 1% PVP-I, 1 mM iodine ($I_2$), PVP-I filtered up (FU; >10 kD) fraction, or PVP-I filtered down (FD; <10 kD) fraction, unless indicated otherwise. **(A)** Cellular metabolism was measured with PrestoBlue. **(B)** Mitochondrial membrane potential was measured with JC-1 using flow cytometry. JC-1 monomers indicate the percentage of cells with disrupted mitochondria. **(C)** Cellular metabolism was measured with PrestoBlue. **(D)** Mitochondrial membrane potential was measured with JC-1 using flow cytometry. **(E)** Mitochondrial function was determined by measuring oxygen consumption rate with Seahorse XF. Dashed lines indicate injection with A: oligomycin; B: FCCP; and C: rotenone/antimycin A. **(F, G)** A549 3D spheroids were stained for calcein AM (viable cells) and Hoechst (nuclei). Spheroids were imaged before and directly after treatment every 3 min for 30 consecutive minutes. Images were quantified for calcein AM expression per cell. Scale bar, 1 mm. **(H)** Mitochondrial membrane potential was measured with JC-1 using flow cytometry. **(I)** Cellular metabolism was measured with PrestoBlue. All graphs show the mean ± SD of three biological replicates. **(A, C, G, I)** Values were normalized to control, which was set to 100% in panels (A, C, G, I). P-values were calculated versus control: *P < 0.05 and ***P < 0.001, **(B, D, I)** t test and **(E, G)** multiple comparisons two-way ANOVA.

PVP-I and $I_2$ (Fig 3A and B). Indeed, ultra-short-term treatment (UST, 5 s) already disrupted cellular metabolism and mitochondrial membrane potential (Figs 3C and S4A). Fine titrations of PVP-I revealed a threshold-controlled instant binary event of unprogrammed cell death, for which we propose the term iodoptosis (Fig S4B and C). As our data showed direct cytoplasmic leaking of calcein AM and loss of the mitochondrial membrane potential, we used cholera toxin B subunit (CTxB) staining to explore whether PVP-I might attack cellular phospholipid bilayers. CTxB binds to the GM1 lipid rafts organizing lipid bilayer membranes (Kenworthy et al, 2021). Upon short-term treatment, membranous lipid rafts became depleted and disorganized. The effect was attributed to $I_2$ (Figs 3D

and S4D). In summary, iodoptosis results from the rapid disintegration of cellular and organellar phospholipid bilayers.

## Clinical concentrations of PVP-I instantly disrupt barrier integrity

Surprisingly, we found that barriers of epithelial, endothelial, and mesothelial cells lose integrity and become leaky upon $I_2$ or PVP-I exposure (Figs 3 and S5). Tight junction failure disrupted tissue integrity, permitting diffusion of excess PVP-I and $I_2$ deep into tissue spheres (Figs 1G and S1E), presumably via intercellular diffusion. The process is conducive to severe systemic toxicity of iodine itself as much as translocation of pathogens and toxins.

**Table 1.  IC$_{50}$ values of PVP-I and I$_2$ in indicated cell lines, as calculated from Fig 1A and C.**

| Cell line | IC$_{50}$ PVP-I (%) | IC$_{50}$ I$_2$ (mM) | Predicted IC$_{50}$ PVP-I if 10% solution (wt/vol) equals 39.4 mM (mM) |
|---|---|---|---|
| HMEC-1 | 0.061 | 0.305 | 0.240 |
| LP-9 | 0.101 | 0.431 | 0.398 |
| A549 | 0.085 | 0.254 | 0.335 |
| RAW264.7 | 0.077 | 0.204 | 0.303 |

PVP-I percentage is transformed into I$_2$ molarity assuming all I$_2$ bound by PVP-I is directly freely available.

**Table 2.  I$_2$ and I$_3^-$ contents present in PVP-I titration series in PBS as measured with UV-Vis spectrometry at 354 nm (I$_3^-$) and 460 nm (I$_2$).**

| PVP-I (%) | I$_2$ content (mM) | I$_3^-$ content (mM) |
|---|---|---|
| 0.250 | 2.292 ± 0.221 | 0.068 ± 0.007 |
| 0.100 | 1.393 ± 0.071 | 0.060 ± 0.0006 |
| 0.050 | 0.455 ± 0.061 | 0.031 ± 0.003 |
| 0.025 | 0.066 ± 0.008 | 0.005 ± 0.0006 |
| 0.010 | 0.029 ± 0.005 | 0.001 ± 0.0001 |

Mean ± SD is shown of three independent experiments.

Lipid rafts assemble cellular tight junctions (Lee et al, 2008). Staining of the zonula occludens-1 (ZO-1) upon short-term treatment with PVP-I revealed their universal and complete abolition (Figs 3E and S5A), mediated by I$_2$ (Fig S5B). Consequently, disruption of tight junctions caused a rapid rise of tissue permeability for both small (4 kD) and large (70 kD) molecules (Figs 3F and S5C and D). Hence, our observations indicate that clinical doses of PVP-I destroy intercellular connections, resulting in leaky linings of exposed airway, peritoneal cavity, and vasculature. They likely explain why experimental "washout" of soiled peritoneum or colon with PVP-I solution produces septicaemia in clinical practice (Whiteside et al, 2005; Song et al, 2018).

Our data provide critical evidence for the reappraisal of PVP-I in clinical practice. Numerous reports have previously raised safety concerns about PVP-I at commonly used concentrations (Ramaswamykanive et al, 2011; Kim et al, 2017; Song et al, 2018; Liang et al, 2020; Vercammen et al, 2021; Fuse et al, 2022). Chronic iodine excess causes thyroid toxicity (Katagiri et al, 2017), yet topical toxicity and systemic overload can have severe and occasionally lethal effects (D'Auria et al, 1990; Glick et al, 1985). They include aspiration pneumonia and pneumonitis (Choi et al, 2014; Hitosugi et al, 2019), iodine burns (Nahlieli et al, 2001), renal failure (Manfro et al, 2002), and sudden cardiovascular collapse (Ramaswamykanive et al, 2011), potentially rescued by timely haemodialysis (Kanakiriya et al, 2003). As such, our findings explain the organ failure or death after systemic I$_2$ overload, and the endotoxaemia and septic shock observed in animal models exposed to PVP-I. Many commonly used clinical regimes are predictably toxic, and self-medication may incur significant harm. Restriction of availability of marketed PVP-I solutions to below 0.1% (wt/vol), meaning the total available I$_2$ should be below 1 mM, combined with dedicated user education, would avert unwarranted toxicity whilst safeguarding, if not even promoting, the many and undisputable benefits of an age-old effective topical antiseptic.

## Materials and Methods

### Cell line culture

HMEC-1 (human endothelial-like cell isolated from the endothelium, kindly provided by Anna Orlov), LP-9 (untransformed human mesothelial cells isolated from peritoneal ascites fluid, kindly provided by Carmela Ricciardelli), A549 (non–small-cell lung carcinoma alveolar basal epithelial cells; ATCC), and RAW264.7 (mouse-derived macrophages; ATCC) cells were cultured according to the standard procedures in culture medium supplemented with 8% foetal bovine serum, penicillin (100 U/ml), and streptomycin (500 $\mu$g/ml; Sigma-Aldrich). HMEC-1 cells were cultured in MCDB131 (Life Technologies) with 10 ng/ml EGF (Promega), and 1 $\mu$g/ml hydrocortisone and 10 mM glutamine (both from Thermo Fisher Scientific). LP-9 cells were cultured in M199 medium and Ham's F12 medium (1:1 ratio; all from Thermo Fisher Scientific) with 10 ng/ml EGF and 0.5 $\mu$g/ml hydrocortisone. A549 cells were cultured in supplemented DMEM/F12, and RAW264.7 cells were cultured in supplemented RPMI 1640 (all from Thermo Fisher Scientific). Cells were monitored for mycoplasma.

### Reagents and treatments

Povidone-iodine (10% PVP-I, Betadine, purchased from the Canberra Hospital Pharmacy), povidone (PVP, European Pharmacopoeia Reference Standard P2660000; Sigma-Aldrich), and 1 mM iodine (Sigma-Aldrich) solution (wt/vol) dissolved in phosphate buffer (17.01 gr. KH$_2$PO$_4$ set to pH 4.0 with 85% phosphoric acid to keep I$_2$ in a stable form) were diluted in PBS into working concentrations, and cells were treated with reagents for indicated periods of time. Glutathione (Sigma-Aldrich) was prepared at 2 mM in PBS. PBS or phosphate buffer was used as a control treatment. Fresh iodine solution was prepared every month, and exact I$_2$ concentration was measured using cuvette-based UV-Vis spectrophotometry at 354 and 460 nm, and/or the leuco crystal violet assay (both assays were validated and equally accurate) (Lambert et al, 1975; Li et al, 2011). For PVP-I filtration experiments, Amicon Ultra-15 centrifugal filter units were used with a pore size of 10 kD (Millipore) according to the manufacturer's instructions. In short, membranes were washed with sterile water and 10 ml of a 1% PVP-I dilution (vol/vol) in PBS was spun down at 4,000$g$ for 30 min. The lower compartment (PVP-I FD) yielded 9.8 ml, whereas the upper compartment yielded 200 $\mu$l (PVP-I FU). The upper compartment was reconstituted in 10 ml of PBS to achieve a similar dilution factor

**Table 3.** $I_2$ and $I_3^-$ contents present in $I_2$ solution titration series in PBS as measured with UV-Vis spectrometry at 354 nm ($I_3^-$) and 460 nm ($I_2$).

| $I_2$ solution (dilution coefficient) | $I_2$ content (mM) | $I_3^-$ content (mM) |
|---|---|---|
| undiluted | 1.021 ± 0.008 | 0.009 ± 0.0004 |
| 1:2 | 0.552 ± 0.007 | 0.005 ± 0.0004 |
| 1:4 | 0.254 ± 0.002 | 0.001 ± 0.00005 |
| 1:10 | 0.128 ± 0.011 | 0.001 ± 0.0004 |
| 1:100 | 0.061 ± 0.009 | 0.001 ± 0.0001 |

Mean ± SD is shown of three independent experiments.

**Table 4.** $I_2$ and $I_3^-$ contents present in PVP-I FU/FD titration series in PBS as measured with UV-Vis spectrometry at 354 nm ($I_3^-$) and 460 nm ($I_2$).

| PVP-I FU (%) | $I_2$ content (mM) | $I_3^-$ content (mM) |
|---|---|---|
| 0.250 | 2.443 ± 0.068 | 0.066 ± 0.002 |
| 0.100 | 1.746 ± 0.055 | 0.061 ± 0.003 |
| 0.050 | 0.472 ± 0.026 | 0.023 ± 0.0009 |
| 0.025 | 0.053 ± 0.017 | 0.003 ± 0.0006 |
| 0.010 | 0.021 ± 0.023 | 0.001 ± 0.0008 |
| PVP-I FD (%) | $I_2$ content (mM) | $I_3^-$ content (mM) |
| 1 | Not detected by UV-Vis | 0.001 ± 0.0001 |

Mean ± SD is shown of three independent experiments.

between both compartments. The solutions of the upper and lower compartments were separately collected and stored at room temperature in the dark.

For all tests, the culture medium was aspirated from cells/spheroids and they were exposed to the indicated treatments diluted in PBS. Afterwards, either treatment was aspirated and a fully supplemented medium was added to the cells or direct analysis was performed.

### Cell viability assays

Cells were seeded at $1 × 10^4$ cells per well in a 96-well plate and left to adhere overnight at 37°C. Cells were treated with the various reagents, washed with PBS, and analysed for cell viability using either PrestoBlue resazurin-based (Thermo Fisher Scientific) or CellTiter-Glo ATP-based (Promega) cell viability reagents according to the manufacturer's instructions.

For live-cell imaging, cells were seeded in glass-bottom black opaque–walled 96-well plates and left to adhere overnight. Cells were stained with 1 $\mu$M calcein AM cell-permeant dye (Thermo Fisher Scientific) for 2 h at 37°C and replaced with 90 $\mu$l serum-free medium. 96-well plates were placed under a Zeiss Axio Observer inverted fluorescence microscope and positioned for imaging. A baseline t = 1 s image was made. Subsequently, at the same time repetitive imaging was started by one person making one image every second for 40 consecutive seconds, whereas another person added 10 $\mu$l of 10 times concentrated therapy to the well without interference with the set-up. Images were analysed and quantified using ImageJ software.

### Spheroid cultures

A549 cells were trypsinized, and $1 × 10^3$ cells per well were plated in 40 $\mu$l fully supplemented medium in 384-well spheroid plates (Corning) using a PerkinElmer JANUS automated liquid handling workstation (PerkinElmer). After 4 d of cultivation, spheroids were stained with 1 $\mu$M calcein AM cell-permeant dye and 1 $\mu$g/ml Hoechst for 3 h at 37°C. Spheroids were washed with PBS and spun down at 79$g$ for 1 min. A time point zero image was made of the spheroids, PBS was aspirated, therapy was added to each well, and spheroids were briefly spun down at 79$g$. Wells were imaged directly afterwards every 3 min for 30 consecutive minutes using the PerkinElmer Opera Phenix high-content imaging microscope (PerkinElmer). For each time point, images were taken of each spheroid on five different planes in the z-axis (i.e., 0, 5, 10, 15, and 20 $\mu$M). Images were analysed using the image analysis algorithms of Harmony software. The amount of total calcein AM per cell per spheroid was calculated by dividing the amount of calcein AM expression per image by the amount of Hoechst-positive nuclei over all five z-axis planes. The spheroid was subsequently divided over an inner region (30% of spheroid) and an outer region (70% of spheroid) to calculate the out/in ratio and approximate speed of diffusion and toxicity.

### Flow cytometry

Adherent cells were stained with the mitochondrial membrane potential stain JC-1 (Thermo Fisher Scientific) at 2 $\mu$M for 30 min at 37°C in T25 flasks. Cells were collected in flow cytometry tubes and treated with indicated reagents for indicated periods of time. For 5-s exposure to treatment, cells were in 90 $\mu$l PBS and 10 $\mu$l 10 times concentrated treatment was added to the tube and directly acquired on the flow cytometer. JC-1 staining was measured on a BD APF LSRII flow cytometer. Samples were gated for single cells and analysed with FlowJo 10 (Tree Star).

### Seahorse

OCRs of cells were measured using the Seahorse XF$^e$96 Extracellular Flux Analyzer (Agilent Technologies) according to the instruction of the Agilent Seahorse XF Cell Mito Stress Kit (103015-100). Cells were seeded in a 96-well XF cell culture microplate a day before the experiment: A549 and HMEC-1 $2 × 10^4$ cells/well, LP-9 $1 × 10^4$ cells/well, and RAW264.7 $5 × 10^4$ cells/well. On the day of the assay, culture media were aspirated. Cells were subjected to respective treatment conditions in PBS, followed by washing of cells and incubating in a non-$CO_2$ incubator for 1 h. OCR was taken at baseline level before injection of the following compounds: 1–1.5 $\mu$mol/litre oligomycin in port A, 0.5 $\mu$mol/litre carbonyl cyanide 4-(tri-fluoromethoxy)phenylhydrazone (FCCP) in port B, and 0.5 $\mu$mol/l rotenone/antimycin A in port C.

### Cell cycle analysis

Plated cells were treated with indicated reagents in PBS for 30 min in an incubator at 37°C, 5% $CO_2$. Subsequently, reagents were

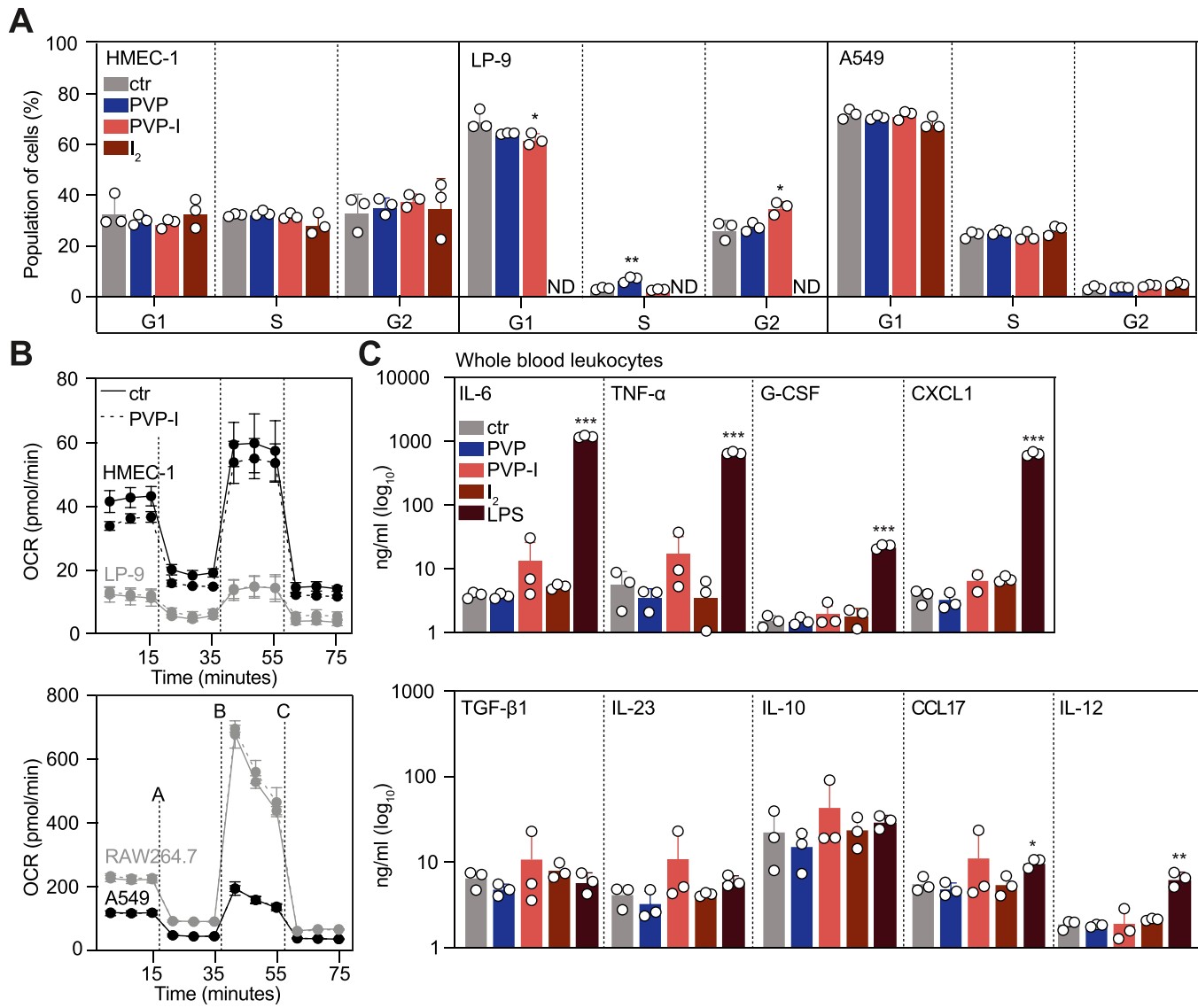

**Figure 2. Long-term exposure to low-dose concentrations of PVP-I does not affect functionality of various cell types.**
HMEC-1, LP-9, A549, and RAW264.7 cells were exposed to low-dose long-term (LT, 30 min) treatment with PBS control, 0.01% PVP, 0.01% PVP-I, or 50 $\mu$M I$_2$. **(A)** 48 h after LT treatment, cell cycle analysis was performed using the Click-iT EdU flow cytometry assay. ND, not determined. **(B)** Directly after LT treatment, mitochondrial function was determined by measuring oxygen consumption rate with Seahorse XF. Dashed lines indicate injection with A: oligomycin; B: FCCP; and C: rotenone/antimycin A. **(C)** Leucocytes were freshly isolated from mouse whole blood. 24 h after LT treatment of leucocytes, cytokine production was measured in the supernatant using the Mouse Cytokine Multiplex Assay with flow cytometry. LPS (100 ng/ml) was used as a positive control. All graphs show the mean ± SD of three biological replicates. *P*-values were calculated versus control: *$P < 0.05$, **$P < 0.01$, and ***$P < 0.001$, **(A, C)** *t* test and **(B)** multiple comparisons two-way ANOVA.

aspirated, and cells were cultured in a fresh supplemented medium for 48 h. Cells were stained with Click-iT EdU Alexa Fluor 488 Flow Cytometry Assay Kit according to the manufacturer's instructions. In short, 10 $\mu$M EdU was added to the culture medium of the adherent cells and left to incubate for 2 h at 37°C. Cells were harvested, fixated, permeabilized, and stained with Alexa Fluor 488 for 30 min at RT in the dark. Cells were counterstained with 7-AAD and acquired on a BD APF LSRII flow cytometer. Samples were gated for single living cells and analysed with FlowJo 10 (Tree Star). LP-9 cells treated with I$_2$ and phosphate buffer control for 30 min had mostly detached after 48 h because of the phosphate buffer and could therefore not be analysed.

**Immune response assay**

Whole blood from five C57BL/6 mice was collected via heart puncture in non-heparinized EDTA-coated capillary tubes. Blood was pooled and treated twice with red blood cell lysis buffer (BioLegend) followed by three washes with PBS. Cells were transferred to 96-well plates and LT-treated in PBS. After 30-min treatment, normal culture medium w/wo lipopolysaccharide (LPS, 100 ng/ml) was added to two lots of triplicates. After 24-h incubation at 37°C, 5% CO$_2$, leucocyte immune response was determined using the LEGENDplex Mouse Macrophage/Microglia Panel (13-plex) according to the manufacturer's instructions using 25 $\mu$l of culture supernatant. Samples were acquired on a BD-LSRII flow cytometer.

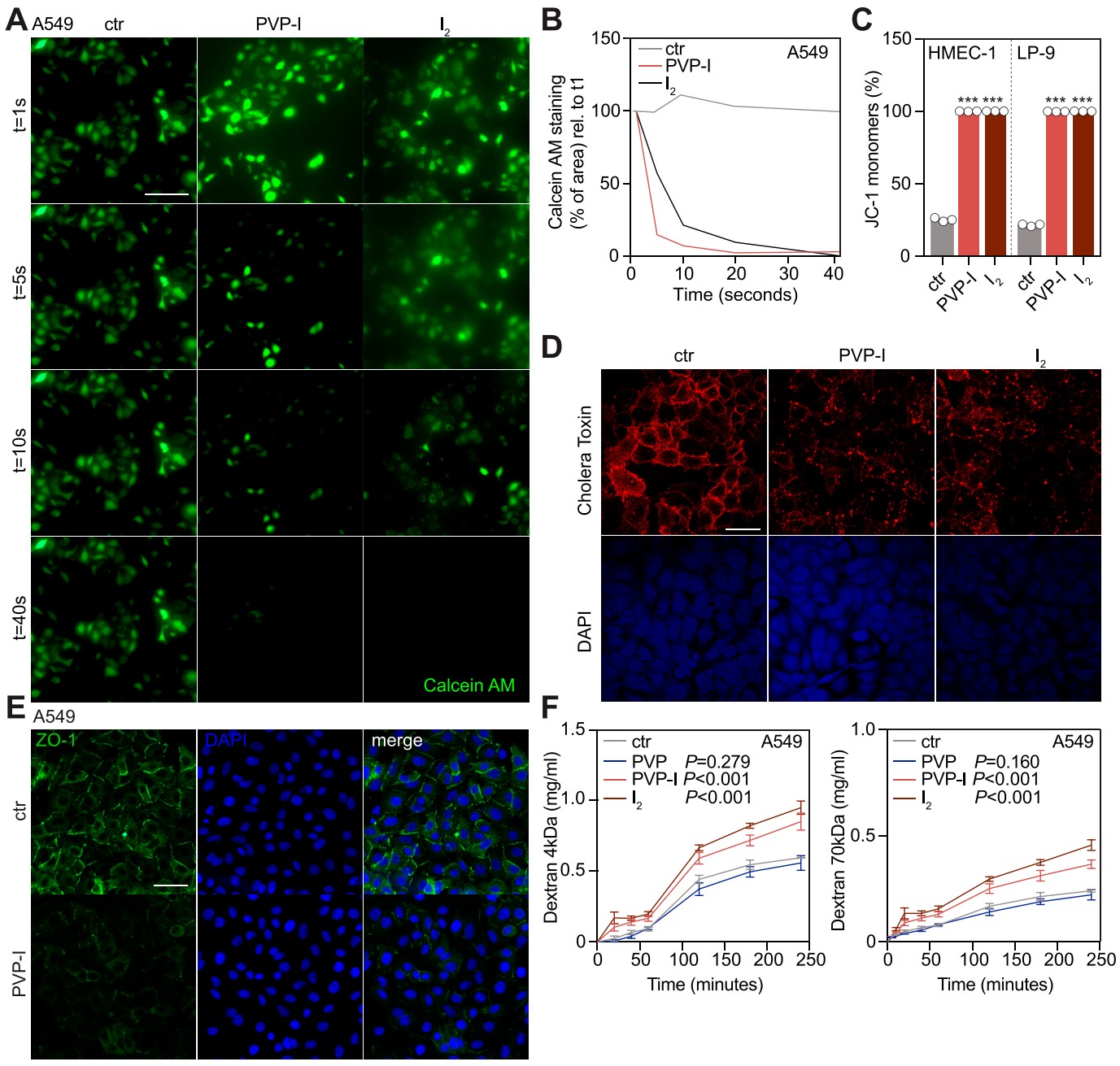

**Figure 3.   PVP-I disrupts the cellular membrane integrity by attacking the cell membrane lipid bilayers.**
Cells were exposed to ST or ultra-short-term (UST, 5 s) treatment with either PBS control, 1% PVP, 1% PVP-I, or 1 mM iodine ($I_2$). **(A, B)** Calcein AM–positive cells were imaged at indicated time points after treatment and quantified as % of area (normalized to t = 1 s). Scale bar, 50 $\mu$m. **(C)** Cells were treated with UST, and mitochondrial membrane potential was measured with JC-1 using flow cytometry after treatment. **(D)** Cell membrane lipid rafts (GM1) were stained with cholera toxin B subunit (red) and DAPI (blue). Scale bar, 50 $\mu$m. **(E)** Cellular tight junctions were stained with zonula occludens-1 (ZO-1, green), and nuclei were stained with DAPI (blue) using immunofluorescence after ST treatment. Scale bar, 50 $\mu$m. **(F)** A549 cells were grown in a Transwell insert until fully confluent and, after ST treatment, 4- and 70-kD dextrans were added to the upper chamber. Aliquots were collected from the lower chamber at fixed time points and analysed for FITC and rhodamine B fluorescence. Graphs in (C, F) show the mean ± SD of three biological replicates. A graph in (B) shows one biological replicate. *P*-values were calculated versus control: \*\*\**P* < 0.001, **(C)** *t* test and **(F)** multiple comparisons two-way ANOVA.

## (Immuno)fluorescence

Cells were seeded on glass coverslips or in glass-bottom black opaque–walled 96-well plates and left to adhere overnight. Wells were treated with indicated reagents for 5 min, washed with PBS, and fixated in 4% PFA. Cells were permeabilized using 1% Triton X-100 (Sigma-Aldrich) in PBS for 10 min and blocked (0.1% Triton X-100 and 5% normal goat serum [Invitrogen] in PBS) for 60 min. ZO-1 monoclonal antibody (33-9100, 1:100; Invitrogen) was incubated overnight at 4°C, and goat anti-mouse Alexa Fluor 488 secondary

antibody (Ab150113, 1:800; Abcam) was incubated for 1 h at room temperature in the dark. Coverslips were mounted with ProLong Gold Antifade Mountant with DAPI (Invitrogen). Images were made on a Zeiss Axio Observer inverted fluorescence microscope.

For cholera toxin B subunit (CTxB) staining, cells were seeded on glass coverslips and after adhesion treated with indicated reagents for 5 min. Cells were then washed with HBSS (Sigma-Aldrich) + 0.5% BSA and stained with CT directly conjugated with Alexa Fluor 647 (Invitrogen) at 1 $\mu$g/ml for 30 min at 4°C in the dark. Cells were washed 5× with cold HBSS + 0.5% BSA, fixed with 4% PFA for 15 min at room temperature, and washed twice with PBS. Coverslips were mounted with ProLong Gold Antifade Mountant with DAPI. Images were acquired on a Leica SP5 confocal microscope.

### Western blot

Cells were treated with 0.1% PVP-I for 2 h, 0.25% PVP-I for 5 min, or PBS control. Samples were washed with cold PBS and directly lysed with lysis buffer and sample loading buffer containing SDS and 100 mM DTT. Immunoblotting was performed as described previously (Feng et al, 2022).

### Transwell permeability assay

A549 cells were seeded into 12-mm Transwell with 0.4-$\mu$M pore polycarbonate membrane inserts (3401; Corning) and left to grow into a differentiated monolayer for 5 d. Medium from the outer chamber was replaced by PBS. Medium from the inner chamber was replaced by indicated therapies in PBS for 5 min. Subsequently, the inner chamber was washed once with PBS and PBS from the inner and outer chambers was replaced by DMEM without phenol red (Thermo Fisher Scientific). 4-kD FITC-labelled (46944; Sigma-Aldrich) and 70-kD rhodamine B–labelled (R9379; Sigma-Aldrich) antibodies were added at 1 mg/ml each to the upper chamber and further incubated at 37°C. Samples of 100 $\mu$l were collected from the outer chamber at fixed time points (5, 10, 20, 40, 60, 120, 180, and 240 min), and the outer chamber was supplemented with 100 $\mu$l fresh DMEM without phenol red after each collection. Samples, including standard curves, were measured in a black opaque walled 96-well plate on a Tecan Infinite 200 PRO plate reader at 485/544 nm for FITC- and 520/590 nm for rhodamine B–labelled dextrans.

### Statistical analysis

A two-sided unpaired $t$ test was used to determine significance. A $P$-value <0.05 was considered statistically significant. Error bars in graphs indicate the SD. Non-linear regression analysis (four parameters) was used to calculate $IC_{50}$ values of reagents. Multiple comparisons two-way ANOVA was used to compare curves in 3D spheroid imaging, seahorse assays, and Transwell permeability assays. All statistical analyses were performed using GraphPad Prism 7.03.

# Supplementary Information

# Acknowledgements

We want to thank Dr. Harpeet Vohra and Michael Devoy within the John Curtin School of Medical Research (JCSMR) at ANU for help with flow cytometry and seahorse assays; Catherine Gillespie (JCSMR) for assistance with confocal imaging; Carmela Ricciardelli, Noor Lokman, and Anna Orlov for providing us with cell lines and help with culturing methods; Jenni Hayward for help with seahorse and JC-1 flow cytometry assays; Jean Capello for help with handling of equipment; and the ANU Centre for Therapeutic Discovery at the JCSMR for their assistance with 3D imaging. We thank the Alexander Pigott Wernher Memorial Trust, London, UK (Charity number: 261362), and the Dutch Research Council (NWO Rubicon grant number 452020215) for funding of this research.

## Author Contributions

A Steins: conceptualization, data curation, formal analysis, validation, investigation, visualization, methodology, project administration, and writing—original draft, review, and editing.
C Carroll: data curation, formal analysis, validation, investigation, visualization, methodology, project administration, and writing—original draft, review, and editing.
FJ Choong: data curation, formal analysis, investigation, methodology, and writing—original draft, review, and editing.
AJ George: resources, data curation, software, visualization, and writing—review and editing.
J-S He: resources, data curation, software, visualization, and writing—review and editing.
KM Parsons: resources, data curation, software, visualization, and writing—review and editing.
S Feng: data curation, formal analysis, and writing—review and editing.
SM Man: data curation, formal analysis, and writing—review and editing.
C Kam: data curation and formal analysis.
LM van Loon: conceptualization, methodology, and writing—review and editing.
P Poh: resources, software, and writing—review and editing.
R Ferreira: resources, software, supervision, and writing—review and editing.
GJ Mann: conceptualization, resources, funding acquisition, and writing—review and editing.
RL Gruen: conceptualization, resources, funding acquisition, and writing—review and editing.
KM Hannan: conceptualization, resources, and methodology.
RD Hannan: conceptualization, resources, supervision, funding acquisition, methodology, and writing—review and editing.
K-M Schulte: conceptualization, resources, formal analysis, supervision, funding acquisition, investigation, methodology, and writing—original draft, review, and editing.

## Conflict of Interest Statement

The authors declare that they have no conflict of interest.

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
