## [Reviewer comments · Life Science Alliance]

Life Science Alliance

Cell Death and Barrier Disruption by Clinically Used Iodine Concentrations

Anne Steins, Christina Carroll, Fui Jiun Choong, Ameer George, Jin-Shu He, Kate Parsons, Shouya Feng, Si Ming Man, Cathelijne Kam, Lex van Loon, Perlita Poh, Rita Ferreira, Graham Mann, Russell Gruen, Katherine Hannan, Ross Hannan, and Klaus-Martin Schulte

DOI: <https://doi.org/10.26508/lsa.202201875>

Corresponding author(s): Klaus-Martin Schulte, Australian National University; Anne Steins, Australian National University; and Klaus-Martin Schulte, Australian National University

Review Timeline:

Submission Date:	2022-12-13
Editorial Decision:	2023-01-23
Revision Received:	2023-02-26
Editorial Decision:	2023-03-06
Revision Received:	2023-03-13
Accepted:	2023-03-14

Scientific Editor: Novella Guidi

Transaction Report:

January 23, 2023

Re: Life Science Alliance manuscript #LSA-2022-01875

Dr. Anne Steins
Australian National University
131 Garran Road
Acton, ACT 2601
Australia

Dear Dr. Steins,

Thank you for submitting your manuscript entitled "Cell Death and Barrier Disruption by Clinically Used Iodine Concentrations" to Life Science Alliance. The manuscript was assessed by expert reviewers, whose comments are appended to this letter. We invite you to submit a revised manuscript addressing the Reviewer comments.

Thank you for this interesting contribution to Life Science Alliance. We are looking forward to receiving your revised manuscript.

Sincerely,

B. MANUSCRIPT ORGANIZATION AND FORMATTING:

Reviewer #1 (Comments to the Authors (Required)):

This is a very valuable study.

It cannot be said that there is no cytotoxicity at concentrations below 0.1% povidone-iodine. 0.1% povidone-iodine has a peak free iodine concentration of 25 ppm, but even 1% and 0.01% povidone-iodine have a bactericidal effect, with a concentration of about 10 ppm.

At high concentrations, the iodine content is high and the bactericidal effect is sustained, but at low concentrations, the iodine content is low and the bactericidal effect is not sustained. 0.1% povidone-iodine peaks, and even 1% and 0.01% show a free iodine concentration of about 10ppm.

However, at 0.01%, the effect does not persist, so there is little cytotoxicity, and at 1%, the effect persists, so there is cytotoxicity.

How about changing the wording to say that such a change occurs with a peak at 0.1% povidone-iodine?

The efficacy of povidone-iodine is estimated to be 0.005-10%¹). 0.005-0.1% is considered to be effective because there is a small amount of damage to membrane proteins in bacteria and normal cells.

1) Van den Broek PJ, et al: Interaction of povidone-iodine compounds, phagocytic cells, and microorganisms. *Antimicrob Agents Chemother.* 1982;22(4):593-7.

Safe concentrations for the retina are reported to be {less than or equal to}0.033%²) safe for cultured corneal endothelial cells {less than or equal to}0.05%³) safe for corneal epithelial cells {less than or equal to}0.12%³) and safe for conjunctival epithelial cells {less than or equal to}0.25%⁴).

2) Shimada H, et al: Evaluation of retinal function and pathology after intravitreal injection of povidone-iodine and polyvinyl alcohol-iodine in *Transl Vis Sci Technol.* 2020;9(5):5.

3) Jiang J, et al: The toxic effect of different concentrations of povidone iodine on the rabbit's cornea. *Cutan Ocul Toxicol.* 2009;28(3),119-124.

4) Swift W, et al: Povidone iodine treatment is deleterious to human ocular surface conjunctival cells in culture. *BMJ Open Ophthalmol.* 2020;5(1):e000545.

Reviewer #2 (Comments to the Authors (Required)):

Review on the manuscript LSA-2022-01875 entitled "Cell death and barrier disruption by clinically used iodine concentrations"
General comments:

This is a highly interesting study on the mechanisms of cytotoxicity of elemental iodine and PVP-iodine. Numerous tests in different cell types on cell viability, metabolism, mitochondria, cell cycle, oxygen consumption, cytokine production, cell membrane integrity, and tight junctions were performed.

The study demonstrates the attack on multiple sites typical for active halogen compounds such as iodine. Additionally, a comparison between a pure iodine solution and a PVP-iodine solution has been performed to investigate if the proposed delayed liberation of iodine from PVP-I has an influence on toxicity. Calculations done from the found cytotoxicity versus concentration of iodine suggest a similar toxicity between both I₂ and PVP-I.

This is reasonable if both solutions have the same oxidation capacity since the toxicity, also in PVP-I, largely relies on the concentration of free iodine (I₂) under normal use conditions.

At this point, the authors could significantly strengthen their conclusions. It would be great if they could provide the results of their measurements of free iodine in their 0.001% to 1% PVP-I solution as well as in their 0.01 - 1 mM iodine solution. This would allow the important direct comparison of molar I₂ in both solutions.

The free iodine content of diluted solutions does not follow a linear function due to the complicated iodine chemistry, particularly for PVP-I. For instance, it has been shown that in a PVP-I solution the free iodine concentration increases when it is diluted from 10% to 0.1% from about 0.01 to 0.1 mM ! Further dilution leads to a decrease in I₂. (for review see for instance [Gottardi W.

Iodine as disinfectant. In: Iodine chemistry and applications. 2015:375-410; page 381 from this book is added at the end of this review]). Specific solutions from different companies may differ in their properties.

This is the reason, why an additional table showing the measured actual iodine content of their solutions at the different applied dilutions (0.01%, 0.1% and 1% PVP-I and 0.01mM, 0.1mM and 1mM iodine Sigma solution) would be very helpful to evaluate if the threshold cytotoxic concentrations in their I2 and PVP-I solutions virtually contained the same I2 concentration or not.

Without these informations, reliable conclusions of the comparison between both solutions are not possible. The authors may try these measurements with the UV method they are familiar with (page 14, lines 264-266). However, there is a caveat. On line 266, they write that both assays they used were validated and equally accurate. The method of reference 50, however, does not distinguish between iodide and iodine etc. From reference 51, this appears possible, but the method may be no standard. If the UV method is insufficient, the reliable standard method would be potentiometric measurement of iodine.

If the authors are unable provide the requested actually measured I2 concentrations, they should mention this as a significant limitation in the discussion.

In this context, also the iodine (I2) concentration in the PVP-I filtered up (FU) and PVP-I filtered down (FD) fraction could be measured. This would probably confirm that most of the I2 is bound to PVP and allow further interpretation of the results.

Page 7, lines 138-145.

The authors are all right to challenge the dogma that PVP-based germicides are not irritating or toxic. It is a pity that companies are not forced to declare the content of free iodine in their preparations, which would allow a better comparison of efficacy and toxicity. We have the same problem with numerous publications...

The decisive point is the concentration of free iodine (I2). This is the statement that can be drawn from references 3, 33, and 34 (and 10 ?). Therefore, these references are not cited in a correct connection. The authors of these publications do not write that PVP-based iodine is not toxic. This should be corrected or other citations used.

Lines 167-169, Figure 2:

The authors mention that they exposed the cells to a "1000-fold lower concentration than used in clinic" and this was "50 microM I2". A saturated iodine solution contains about 1.3 mM iodine [R.W. Ramette, R.W.J. Sandford, Thermodynamics of iodine solubility and triiodide ion formation in water and in deuterium oxide, J. Am. Chem. Soc. 87(22) (1965) 5001-5005] so that this statement is not correct. Otherwise preparations would contain 50 mM iodine. As mentioned above, I2 concentration does not follow in a linear way in diluted solutions (it can even increase despite dilution under certain conditions).

Page 8, prolonged exposure to low dose. Just a few comments that might be useful for discussion:

It is clear that exposure to lower concentrations of iodine (similar to virtually all antiseptics) exerts lower cytotoxicity than exposure to higher concentrations.

The decisive question is to find the optimal compromise between tolerability and efficacy in vivo.

Concentrations which are no more cytotoxic in vitro hardly have sufficient antimicrobial efficacy already in vitro.

Therefore, the application concentration of antiseptics in vivo exceeds the minimum cytotoxic concentration in vitro. Otherwise, they have no efficacy. Luckily, the tissue in vivo generally tolerates significantly higher concentrations of antiseptics than the cell culture in vitro.

Moreover, in this context it must be taken into account that reaction of iodine with exudate, body fluid etc. reduces the concentration of free iodine for the microbicidal action in vivo.

All these considerations demonstrate that it is a complex procedure to find the optimal in-vivo dosing regimen from in-vitro testing.

They do not curtail the merits of the present study in disclosing mechanisms of toxicity, which is important for this process of dose finding.

Materials and Methods:

It is not described in all assays (e.g. cell viability assays, page 15) if the test reagents were incubated with cells in plain PBS or in cell culture medium. This is of importance because of reaction of iodine with the medium. Please add either a general statement that all tests have been performed in PBS with washed cells or clearly describe the way it was done for each test.

Line 45: I suggest to replace "caustic" by the more specific "oxidative damage"

Line 98: remove "animals"

Fig. 1 A and C: use other colours for HMEC-1 and LP-9 since they hardly can be distinguished in the figures.

Fig. 1 in general: No statistics are provided for panels A, C, E, G, and I.

It should be stated in the legend that P values were calculated versus controls.

Fig. 2: Statistics for panels B and C are missing.

It should be stated in the legend that P values were calculated versus controls.

Fig. 3B: Would use another colour for I2 to make it better distinguishable.

Statistics are missing for 3F.

It should be stated in the legend that P values were calculated versus controls.

Line 382: ... and with from ... This is confusing. There seems to be something missing.

Suppl Fig. 1:

Use different colors in panels B, C, and F for single curves.

Statistics are missing.

line 543: F instead of G for the panel legend.

Suppl Fig. 3:

Statistics are missing in panels A: Was 5 sec PVP-I significant versus control ? What does ND mean ? Not determined / done or not detectable ?

Gottardi W. Iodine as disinfectant (Chapter 20). In: Iodine chemistry and applications. 2015:375-410. Editor: Kaiho, T. Publisher: John Wiley & Sons, Inc., ISBN: 978-1-118-46629-2 (cloth)

Reviewer #1 (Comments to the Authors (Required)):

This is a very valuable study.

We thank reviewer #1 for the compliment on our work.

It cannot be said that there is no cytotoxicity at concentrations below 0.1% povidone-iodine. 0.1% povidone-iodine has a peak free iodine concentration of 25 ppm, but even 1% and 0.01% povidone-iodine have a bactericidal effect, with a concentration of about 10 ppm.

We thank reviewer #1 for this comment and have amended our manuscript to 'eukaryotic cytotoxicity'.

At high concentrations, the iodine content is high and the bactericidal effect is sustained, but at low concentrations, the iodine content is low and the bactericidal effect is not sustained. 0.1% povidone-iodine peaks, and even 1% and 0.01% show a free iodine concentration of about 10ppm.

However, at 0.01%, the effect does not persist, so there is little cytotoxicity, and at 1%, the effect persists, so there is cytotoxicity.

How about changing the wording to say that such a change occurs with a peak at 0.1% povidone-iodine?

We thank reviewer #1 for this comment and have integrated this and the comments below in the 'Results and discussion' section, paragraph 4, as follows;

Bactericidal efficacy of PVP-I solutions is observed in the dilution range of 0.005%- 10%, with free I₂ concentrations peaking at a concentration of 0.1%. Former authors found only a range between 0.005-0.1% effective, as it caused only minor membrane damage (Van den Broek, Buys et al., 1982). Others found the safe concentration range to vary with cell type, such as <0.033% for the retina (Shimada et al., 2020), <0.05% for corneal endothelial cells (Jiang, Wu et al., 2009), and <0.05% for corneal epithelial cells (Swift, Bair et al., 2020).

The efficacy of povidone-iodine is estimated to be 0.005-10%1). 0.005-0.1% is considered to be effective because there is a small amount of damage to membrane proteins in bacteria and normal cells.

1)Van den Broek PJ, et al: Interaction of povidone-iodine compounds, phagocytic cells, and microorganisms. *Antimicrob Agents Chemother.* 1982;22(4):593-7.

See above

Safe concentrations for the retina are reported to be {less than or equal to}0.033%:2) safe for cultured corneal endothelial cells {less than or equal to}0.05%3) safe for corneal epithelial cells {less than or equal to}0.12%3) and safe for conjunctival epithelial cells {less than or equal to}0.25%4).

2)Shimada H, et al: Evaluation of retinal function and pathology after intravitreal injection of povidone-iodine and polyvinyl alcohol-iodine in *Transl Vis Sci Technol.* 2020;9(5):5.

3) Jiang J, et al: The toxic effect of different concentrations of povidone iodine on the rabbit's cornea. *Cutan Ocul Toxicol.* 2009;28(3),119-124.

4) Swift W, et al: Povidone iodine treatment is deleterious to human ocular surface conjunctival cells in culture. *BMJ Open Ophthalmol.* 2020;5(1):e000545.

See above

Reviewer #2 (Comments to the Authors (Required)):

Review on the manuscript LSA-2022-01875 entitled "Cell death and barrier disruption by

clinically used iodine concentrations"

General comments:

This is a highly interesting study on the mechanisms of cytotoxicity of elemental iodine and PVP-iodine. Numerous tests in different cell types on cell viability, metabolism, mitochondria, cell cycle, oxygen consumption, cytokine production, cell membrane integrity, and tight junctions were performed.

We thank reviewer #2 for the kind words.

The study demonstrates the attack on multiple sites typical for active halogen compounds such as iodine. Additionally, a comparison between a pure iodine solution and a PVP-iodine solution has been performed to investigate if the proposed delayed liberation of iodine from PVP-I has an influence on toxicity. Calculations done from the found cytotoxicity versus concentration of iodine suggest a similar toxicity between both I₂ and PVP-I.

This is reasonable if both solutions have the same oxidation capacity since the toxicity, also in PVP-I, largely relies on the concentration of free iodine (I₂) under normal use conditions. At this point, the authors could significantly strengthen their conclusions.

It would be great if they could provide the results of their measurements of free iodine in their 0.001% to 1% PVP-I solution as well as in their 0.01 - 1 mM iodine solution. This would allow the important direct comparison of molar I₂ in both solutions.

The free iodine content of diluted solutions does not follow a linear function due to the complicated iodine chemistry, particularly for PVP-I. For instance, it has been shown that in a PVP-I solution the free iodine concentration increases when it is diluted from 10% to 0.1% from about 0.01 to 0.1 mM ! Further dilution leads to a decrease in I₂. (for review see for instance [Gottardi W. Iodine as disinfectant. In: Iodine chemistry and applications. 2015:375-410; page 381 from this book is added at the end of this review]). Specific solutions from different companies may differ in their properties. This is the reason, why an additional table showing the measured actual iodine content of their solutions at the different applied dilutions (0.01%, 0.1% and 1% PVP-I and 0.01mM, 0.1mM and 1mM iodine Sigma solution) would be very helpful to evaluate if the threshold cytotoxic concentrations in their I₂ and PVP-I solutions virtually contained the same I₂ concentration or not.

Without these informations, reliable conclusions of the comparison between both solutions are not possible. The authors may try these measurements with the UV method they are familiar with (page 14, lines 264-266). However, there is a caveat. On line 266, they write that both assays they used were validated and equally accurate. The method of reference 50, however, does not distinguish between iodide and iodine etc. From reference 51, this appears possible, but the method may be no standard. If the UV method is insufficient, the

reliable standard method would be potentiometric measurement of iodine.

If the authors are unable provide the requested actually measured I₂ concentrations, they should mention this as a significant limitation in the discussion.

We thank reviewer #2 for the comments and agree it is a valuable addition to include these comparisons in the paper. We have measured the iodine and iodide concentrations using UV-Vis spectrophotometry. PVP-I was measured at 0.01, 0.025, 0.05, 0.1 and 0.25%. Unfortunately, 1% PVP-I concentration was above detection limit of the apparatus and could not be measured with UV-Vis. Iodine solution from Sigma was measured in the following dilutions; 1:100, 1:10, 1:4, 1:2 and undiluted, which roughly equalled 0.06mM, 0.1mM, 0.25mM, 0.55mM and 1mM. Unfortunately, 0.01mM was below detection limit of the apparatus and could not be measured with UV-Vis spectrophotometry.

We indeed find that concentrations of I₂ are in the same range at 0.5mM Iodine solution from Sigma and 0.05% PVP-I. We have added the results of these experiments in Table 2 and 3, and supplementary Fig S2, and in the text of the 'Results and discussion' section, paragraph 2.

In this context, also the iodine (I₂) concentration in the PVP-I filtered up (FU) and PVP-I filtered down (FD) fraction could be measured. This would probably confirm that most of the I₂ is bound to PVP and allow further interpretation of the results.

We thank reviewer #2 for this comment and have added the measurements with UV-Vis spectrophotometry of the PVP-I FU and FD to the results section in Table 4 and supplementary Fig S2, and in the text of the 'Results and discussion' section, paragraph 2.

Page 7, lines 138-145.

The authors are all right to challenge the dogma that PVP-based germicides are not irritating or toxic. It is a pity that companies are not forced to declare the content of free iodine in their preparations, which would allow a better comparison of efficacy and toxicity. We have the same problem with numerous publications...

The decisive point is the concentration of free iodine (I₂). This is the statement that can be drawn from references 3, 33, and 34 (and 10 ?). Therefore, these references are not cited in a correct connection. The authors of these publications do not write that PVP-based iodine is not toxic. This should be corrected or other citations used.

We thank reviewer #2 for the comment and have made our manuscript more explicit to avoid any misunderstanding by amending the following in the 'Results and discussion' section, paragraph 3;

'Our observations challenge the entrenched dogma that PVP-based "*germicides containing a high level of molecular iodine are not irritating or toxic*" (Hickey et al., 1997). Other studies

show slow release and fast dissociation of I₂ in pure aqueous media thereby reducing its toxicity to surrounding tissues (Gottardi, 1985, Gottardi, 1999, Gottardi & Nagl, 2019, Zamora, 1986). We propose that the toxicity of I₂ storing polymers is not dependent on the concentration of I₂ liberated at any particular moment. Rather, toxicity depends on the total amount of I₂ which can be liberated. This is due to the fact, that the liberation of I₂ from such polymers is more or less instant as I₂ out reacts with halogen targets as soon as liberated from storage, thereby affording a near-instantaneous halogen challenge to any available targets.'

Lines 167-169, Figure 2: The authors mention that they exposed the cells to a "1000-fold lower concentration than used in clinic" and this was "50 microM I₂". A saturated iodine solution contains about 1.3 mM iodine [R.W. Ramette, R.W.J. Sandford, Thermodynamics of iodine solubility and triiodide ion formation in water and in deuterium oxide, J. Am. Chem. Soc. 87(22) (1965) 5001-5005] so that this statement is not correct. Otherwise preparations would contain 50 mM iodine. As mentioned above, I₂ concentration does not follow in a linear way in diluted solutions (it can even increase despite dilution under certain conditions).

We agree with reviewer #2 that this is incorrect. Only the PVP-I at 0.01% is a 1000-fold lower than used in clinic. I₂ solution was only diluted approximately 50-fold. We have removed that claim from the figure legend.

Page 8, prolonged exposure to low dose. Just a few comments that might be useful for discussion: It is clear that exposure to lower concentrations of iodine (similar to virtually all antiseptics) exerts lower cytotoxicity than exposure to higher concentrations. The decisive question is to find the optimal compromise between tolerability and efficacy in vivo. Concentrations which are no more cytotoxic in vitro hardly have sufficient antimicrobial efficacy already in vitro. Therefore, the application concentration of antiseptics in vivo exceeds the minimum cytotoxic concentration in vitro. Otherwise, they have no efficacy. Luckily, the tissue in vivo generally tolerates significantly higher concentrations of antiseptics than the cell culture in vitro. Moreover, in this context it must be taken into account that reaction of iodine with exudate, body fluid etc. reduces the concentration of free iodine for the microbicidal action in vivo. All these considerations demonstrate that it is a complex procedure to find the optimal in-vivo dosing regimen from in-vitro testing. They do not curtail the merits of the present study in disclosing mechanisms of toxicity, which is important for this process of dose finding.

We thank the reviewer for the comments and agree with all said above.

Materials and Methods:

It is not described in all assays (e.g. cell viability assays, page 15) if the test reagents were incubated with cells in plain PBS or in cell culture medium. This is of importance because of reaction of iodine with the medium. Please add either a general statement that all tests have been performed in PBS with washed cells or clearly describe the way it was done for each test.

We agree with reviewer #2 that iodine out reacts with the components of the medium – and therefore performed all exposures in PBS. We have added a general statement to the methods section, at the end of paragraph 'reagents and treatments', that all tests were performed in PBS.

Line 45: I suggest to replace "caustic" by the more specific "oxidative damage"

We have shortened the abstract considerably per journal guidelines instructions and have removed the sentence altogether.

Line 98: remove "animals"

We have removed 'animals'.

Fig. 1 A and C: use other colours for HMEC-1 and LP-9 since they hardly can be distinguished in the figures.

We have changed the colours of indicated cell lines in Fig 1A and C.

Fig. 1 in general: No statistics are provided for panels A, C, E, G, and I.

It should be stated in the legend that P values were calculated versus controls.

We have not performed statistical analysis for panel A and C as we found no relevance in comparing inter cell-line cytotoxicity. Panel G had P-values (not asterisks) already indicated in the figure panel, next to the legend. Statistical values are added to panel E and I and we have added that P-values were calculated versus controls in the figure legend.

Fig. 2: Statistics for panels B and C are missing.

It should be stated in the legend that P values were calculated versus controls.

For panel B statistics were performed but there were no statistically significant differences between control and PVP-I treated cells of each cell line (the filled and dashed line, which mostly overlap), hence no statistical values or asterisks are given. Statistics are added to panel C and we have added that P-values were calculated versus controls in the figure legend.

Fig. 3B: Would use another colour for I₂ to make it better distinguishable.

Statistics are missing for 3F.

It should be stated in the legend that P values were calculated versus controls.

We have adjusted the colour of I₂ in panel 3B. Panel 3F had P-values (not asterisks) already indicated in the figure panel, next to the legend.

Line 382: ... and with from ... This is confusing. There seems to be something missing.

We have adjusted the text to; 'Medium from the outer chamber was replaced by PBS.

Medium from the inner chamber was replaced by indicated therapies in PBS for 5 minutes.'

Suppl Fig. 1:

Use different colors in panels B, C, and F for single curves.

Statistics are missing.

We have changed the colours of the lines in the panels and added statistics to the panels.

line 543: F instead of G for the panel legend.

We have moved supplementary Figure 1 panel F to supplementary Fig S2 panel D

Suppl Fig. 3:

Statistics are missing in panels A: Was 5 sec PVP-I significant versus control ? What does ND mean ? Not determined / done or not detectable ?

We have added the statistics to panel A. ND means; no viable cells were detected. To make this clearer we have removed ND and added the individual datapoints.

Gottardi W. Iodine as disinfectant (Chapter 20). In: Iodine chemistry and applications.

2015:375-410. Editor: Kaiho, T. Publisher: John Wiley & Sons, Inc.,

ISBN: 978-1-118-46629-2 (cloth)

March 6, 2023

RE: Life Science Alliance Manuscript #LSA-2022-01875R

Dr. Anne Steins
Australian National University
131 Garran Road
Acton, ACT 2601
Australia

Dear Dr. Steins,

Thank you for submitting your revised manuscript entitled "Cell Death and Barrier Disruption by Clinically Used Iodine Concentrations". We would be happy to publish your paper in Life Science Alliance pending final revisions necessary to meet our formatting guidelines.

- please address the final remaining Reviewer 2's points
- please consult our manuscript preparation guidelines <https://www.life-science-alliance.org/manuscript-prep> and make sure your manuscript sections are in the correct order
- please add the supplementary figure legends to the main manuscript text
- please add the Twitter handle of your host institute/organization as well as your own or/and one of the authors in our system
- please use the [10 author names, et al.] format in your references (i.e. limit the author names to the first 10)

Figure Check:

- please add a scale bar to the bottom left figure in S5A

A. FINAL FILES:

B. MANUSCRIPT ORGANIZATION AND FORMATTING:

Sincerely,

Reviewer #1 (Comments to the Authors (Required)):

Research on povidone-iodine is advancing rapidly. It is a novel study, and this paper provides the reader with new ideas.

Reviewer #2 (Comments to the Authors (Required)):

The authors have made the requested adjustments.

Above all, they performed and presented, respectively, their measurements of iodine content of the test solutions, which provides a clear basis for the study.

It is now obvious that the spectrophotometric method measures the total available iodine of the solutions, which can be regarded as a decisive parameter for toxicity according to the results of the study. The authors could stress this in the abstract and in the conclusion and ideally add to the approximate threshold concentration of 0.1% PVP-I that the total available iodine should roughly be lower than 1 mM. (In the end, it would be good if companies were forced to provide the iodine concentration to make their preparations comparable also regarding toxicity).

One small remaining point:

The citations for the meanwhile changed sentence "Other studies show slow release and fast dissociation of I₂ in pure aqueous media thereby reducing its toxicity to surrounding tissues" are still not correct, at least for the Gottardi references. To my knowledge, Gottardi has never worked with tissue or cell cultures and iodine. There is no hint in these publications, too. A surely correct citation would be for a sentence like "Other studies investigated the chemical behaviour of different iodine solutions and scrutinized methods to differentiate between iodine species occurring in such solutions".

March 14, 2023

RE: Life Science Alliance Manuscript #LSA-2022-01875RR

Prof. Klaus-Martin Schulte
Australian National University
John Curtin School of Medical Research
131 Garran Road
Acton, ACT 2600
Australia

Dear Dr. Schulte,

Thank you for submitting your Research Article entitled "Cell Death and Barrier Disruption by Clinically Used Iodine Concentrations". It is a pleasure to let you know that your manuscript is now accepted for publication in Life Science Alliance. Congratulations on this interesting work.

DISTRIBUTION OF MATERIALS:

Again, congratulations on a very nice paper. I hope you found the review process to be constructive and are pleased with how the manuscript was handled editorially. We look forward to future exciting submissions from your lab.

Sincerely,
